# Old Methods, New Insights: Reviewing Concepts on the Ecology of Trypanosomatids and *Bodo* sp. by Improving Conventional Diagnostic Tools

**DOI:** 10.3390/pathogens12010071

**Published:** 2023-01-02

**Authors:** Fernanda Moreira Alves, Cristiane Varella Lisboa, Maria Augusta Dario, Roberto Leonan Morim Novaes, Liliani Marilia Tiepolo, Ricardo Moratelli, Ana Maria Jansen

**Affiliations:** 1Laboratory of Trypanosomatid Biology, Oswaldo Cruz Institute, Oswaldo Cruz Foundation, Rio de Janeiro 21040-900, Brazil; 2Oswaldo Cruz Foundation, Fiocruz Atlantic Forest, Rio de Janeiro 22713-375, Brazil; 3Laboratory for Analysis and Monitoring of the Atlantic Forest, Coastal Campus, Federal University of Paraná, Matinhos 83260-000, Brazil

**Keywords:** *Trypanosoma*, *Bodo*, diagnostic tools, hemoculture, blood clot, selection, mixed infections, bats

## Abstract

Mixed infections by different *Trypanosoma* species or genotypes are a common and puzzling phenomenon. Therefore, it is critical to refine the diagnostic techniques and to understand to what extent these methods detect trypanosomes. We aimed to develop an accessible strategy to enhance the sensitivity of the hemoculture, as well as to understand the limitations of the hemoculture and the blood clot as a source of parasitic DNA. We investigated trypanosomatid infections in 472 bats by molecular characterization (18S rDNA gene) of the DNA obtained from the blood clot and, innovatively, from three hemoculture sample types: the amplified flagellates (“isolate”), the pellet of the culture harvested in its very initial growth stage (“first aliquot”), and the pellet of non-grown cultures with failure of amplification (“sediment”). We compared (a) the characterization of the flagellates obtained by first aliquots and isolates; and (b) the performance of the hemoculture and blood clot for trypanosomatid detection. We observed: (i) a putative new species of *Bodo* in *Artibeus lituratus*; (ii) the potential of *Trypanosoma cruzi* selection in the hemoculture; (iii) that the first aliquots and sediments overcome the selective pressure of the hemoculture; and (iv) that the blood clot technique performs better than the hemoculture. However, combining these methods enhances the detection of single and mixed infections.

## 1. Introduction

The class Kinetoplastea comprises flagellated protists characterized by the presence of the kinetoplast, a highly condensed organelle consisting of mitochondrial DNA [1]. From the most basal to the derived evolutionary lineage, this class includes the orders Prokinetoplastida, Neobodonida, Parabodonida, Eubodonida, and Trypanosomatida [2,3,4]. Members of Prokinetoplastida and Trypanosomatida are parasites (but see [5]). Neobodonida and Parabodonida are mainly composed of free-living beings, with a few exceptions of parasitic species. Organisms of the order Eubodonida—uniquely represented by the genus *Bodo* (family Bodonidae)—are known, to date, to be free-living heterotrophs, found globally in aquatic environments [3,6,7].

The level of knowledge about kinetoplastids presents huge discrepancies. Due to their impact on the economy and human health, most attention is given to a few members of the family Trypanosomatidae [8]—the only representative of the order Trypanosomatida. Trypanosomatids include monoxenous genera, mainly found in the digestive tract of insects, and heteroxenous genera [4,9]. The genus *Trypanosoma* comprises nearly 500 species of dixenous hemoparasites, adapted to all classes of vertebrates and some hematophagous invertebrates worldwide [9,10]. One of the main studied organisms is *Trypanosoma cruzi*, the causative agent of Chagas disease [11]. This species shows extraordinary genetic and phenotypic heterogeneity; thus, it is currently divided into seven Discrete Typing Units (DTU), from TcI to TcVI and TcBat [12,13,14,15]. In contrast, the diversity and the biological and ecological aspects of most of the trypanosome species are poorly known [16,17]. Indeed, increasing findings of genetic sequences of putative undescribed novel species, called Molecular Operational Taxonomic Units (MOTUs, see [18]), expose the knowledge gaps regarding the diversity of this genus.

Considering the different classifications within the genus *Trypanosoma*, herein, we used the term “taxon” to encompass the trypanosome species, the DTUs of *T. cruzi,* and the MOTUs.

Bats (Order Chiroptera) are important hosts of a high diversity of trypanosomes, and novel species and MOTUs have recently been discovered in these hosts [16]. For instance, the MOTU *Trypanosoma* sp. Neobat 4 [19] was recognized as a successful *Carollia* bat parasite [17]. Furthermore, fresh reports of other kinetoplastids in bats evidence that we are only beginning to understand such interactions. Bats were found to be infected with monoxenous trypanosomatids, including *Crithidia mellificae* [17,20,21]. Although the infection cannot be confirmed, there are reports of *Bodo saltans* genetic material in bat blood [22] and in their ectoparasites [23].

One of the main methods for diagnosing *Trypanosoma* in mammals consists of the steps of detection, amplification, and isolation of the blood flagellates in axenic culture media [24,25]. Although the hemoculture (HC) allows the study of biological and biochemical properties of the microorganism (e.g., [26,27]), this technique may not reflect the real picture of the hemoparasite community in the host due to its low sensitivity [28]. The HC yields false-negative results in cases of subpatent infections (low parasitemia) and infections by species unable to grow in the artificial media [17]. Furthermore, this technique may act as a filter, selecting certain species or genotypes [29,30,31,32,33]. To overcome these issues, Rodrigues et al. [19] demonstrated that the blood clot (BC) is a valuable source of *Trypanosoma* DNA. These authors observed detection of cultivable and non-cultivable trypanosomes with high sensitivity in wild mammals with negative HC.

Powerful sequencing tools revealed a high prevalence of mixed infections by different trypanosome species or genotypes [16,34,35,36], and interactions among coexisting parasites may modulate the outcomes of infection in the individual host [37]. However, since the true diversity of the genus *Trypanosoma* is unknown, we are incapable of estimating possible interactions or their effects on the host. Therefore, it is critical to refine the diagnostic techniques and to understand to what extent these methods detect trypanosomatids for a proper interpretation of the results, especially in cases of host–parasite interactions which are still misunderstood, such as bats-trypanosomatids.

Our goals were: (1) to improve the detection of trypanosomatids by the HC; and (2) to understand the limitations of the HC and BC techniques by investigating the infection of these parasites in bats. Unlike Rodrigues et al. [19], we evaluated animals with positive and negative hemocultures. We hypothesized that isolation by the HC underestimates the parasite diversity, and that the BC technique is the gold standard for diagnosing trypanosomatid infection.

## 2. Materials and Methods

### 2.1. Dataset

We evaluated the trypanosomatid infection in 472 bats, belonging to 39 species and 22 genera, from seven scattered localities across Brazil (Figure 1). Part of our dataset was taken from [17,20,21,38,39] (Table 1).

### 2.2. Bat Sampling

Bats were captured using ground-level mist nets (9 × 3 m, 20 mm mesh) and taken to the field laboratory. The animals were anaesthetized via intramuscular injection (ketamine chloridrathe 10% and acepromazine 1%) and submitted to a careful asepsis using bactericidal soap, iodized alcohol, and 70% alcohol, prior to blood collection via intracardiac puncture in an aseptic environment with a camp stove.

### 2.3. Hemoculture and Blood Clot Collection

We evaluated the trypanosomatid infection in bats by molecular characterization of the DNA obtained from three HC sample types (isolate, sediment, and first aliquot) and from the BC (Figure 2).

For the HC, bat blood samples were cultured in two tubes containing NNN/LIT (Novy-MacNeal-Nicolle/liver infusion tryptose overlay) and NNN/Schneider’s Insect medium overlay (0.2–0.4 mL in each tube), respectively. The tubes were incubated at 28 °C and analyzed at the Laboratory of Trypanosomatid Biology every two weeks for up to four months. The cultures with sustainable growth were amplified until the stationary phase, cryopreserved, and deposited in the Coleção de *Trypanosoma* de Mamíferos Silvestres, Domésticos e Vetores, Coltryp/Fiocruz. These samples are called “isolates”. The liquid phase of the cultures that showed failure of growth was centrifuged at 1180× *g* for 15 min, and the supernatant was discarded. These pellets are called “sediments”. To evaluate a putative selection during parasite growth, we compared the culture before and after amplification. For this, we collected a portion of the liquid phase of positive hemocultures (*n* = 12) once the flagellates were detected (i.e., in their very initial growth stage), centrifuged it at 1180× *g* for 15 min, and discarded the supernatant. These pellets are called “first aliquot”. The obtention of the HC samples is illustrated in Figure 2.

The BC was obtained by centrifugation of the blood samples at 1180× *g* for 15 min, followed by separation of the supernatant using sterile tips. The samples were stored at −20 °C with absolute ethanol (1:1).

### 2.4. Molecular Diagnosis

The technique of DNA extraction was performed according to the sample type (Table 2). The DNA samples were resuspended in Tris-EDTA buffer (10 mM Tris-HCl pH 7.4; 1 mM EDTA pH 8.0), and the concentration and purity were quantified (OD260/OD280 ratio) using the BioPhotometer^®^ (Eppendorf, Hamburg, Germany).

The nested-PCR targeting the 18S small subunit of the ribosomal gene (SSU rDNA) was performed using GoTaq^®^ Green Master Mix (Promega, Madison, WI, EUA) and the external primers TRY927F (5′ GAAACAAGAAACACGGGAG 3′) and TRY927R (5′ CTACTGGGCAGC TTGGA 3′), at a final volume of 25 µL [41]. The concentrations of the chemical components were adjusted according to the sample type (Table 2). We used 2 to 5 µL of the DNA samples to achieve a maximum of 200 µg of DNA per reaction. For the second round, the PCR products, diluted at 1:10 in ultrapure sterile water, were used as templates, following the same protocol as in the first round with the internal primers SSU561F (5′ TGGGATAACAAAGGAGCA 3′) and SSU561R (5′ CTGAGACTGTAACCTCAAAGC 3′). The amplification was performed on a Swift™ MiniPro Thermal Cycler (Esco Scientific, Singapore) with the following cycling conditions: 94 °C/3 min; 30 cycles at 94 °C/30 s, 55 °C/60 s, and 72 °C/90 s; and 72 °C/10 min. To confirm the results of some isolate samples (Appendix A), we performed a PCR of another region of the SSU rDNA with the primers 609F (5′ CACCCGCGGTAATTCCAGC 3′) and 706R (5′ CTGAGACTGTAACCTCAA 3′) submitted to 30 cycles as follows: 1 min at 94 °C, 2 min at 48 °C, and 2 min at 72 °C (with an initial cycle of 3 min at 94 °C and a final cycle of 10 min at 72 °C) [42]. Ultrapure sterile water and *T. cruzi* DNA from positive hemoculture were used, respectively, as negative and positive controls in all reactions.

The products derived from the second-round reaction were visualized on 1.5–2.0% agarose gels stained with GelRed (Biotium, Fremont, CA, USA). The PCR from the blood clot sample of one bat (Voucher LBT 7097) resulted in double DNA bands (ca. 550 and 600 bp). The bands were collected one by one with sterile scalpel blades, and were separately analyzed. The DNA of each band was purified with Illustra GFX PCR DNA and Gel Band Purification Kit (GE Healthcare Life Sciences, Little Chalfont, Buckinghamshire, UK), according to the manufacturer’s protocol.

The DNA of the PCR products derived from the isolate samples was purified using the Illustra GFX PCR DNA and Gel Band Purification Kit, according to the manufacturer’s protocol. The PCR products derived from the first aliquot, the sediment, and the blood clot were directly sequenced, since the column-based purification results in loss of DNA (according to the manufacturer’s protocol), and these sample types show low DNA concentration.

Both forward and reverse fragment strands of the purified DNA samples and the non-purified PCR products were sequenced at the Oswaldo Cruz Foundation Sequencing Platform facility (PDTIS/FIOCRUZ, Rio de Janeiro, Brazil). The samples were subjected to fluorescent dye-terminator cycle sequencing reactions with the ABI 3730 BigDye Terminator v3.1 Cycle Sequencing Ready Reaction Kit (Applied Biosystems, Waltham, MA, USA) and run on an ABI 3730 automated sequencer (Applied Biosystems). Primer concentrations are shown in Table 2.

The chromatograms of both strands were inspected and manually edited using SeqMan Lasergene v.7.0 (DNASTAR Inc., Madison, WI, EUA). The consensus sequences were compared against the GenBank database with the BLASTn algorithm [43], considering 99.0% as the identity and coverage cut-off. Due to the high similarity among the MOTUs *T*. spp-Neobats [19], the identity cut-off for this group was set to 99.5%.

### 2.5. Phylogenetic Analysis

The sequence obtained from the BC sample of LBT 10097 was aligned to other kinetoplastid sequences retrieved from the GenBank database using the algorithm L-INS-I, available in MAFFT v.7.0 software (Japan) [44]. The alignment was inspected and manually edited on MEGA7 [45]. Maximum likelihood (ML) estimation and Bayesian inference (BI) were performed. For each analysis, the best base substitution models were chosen according to the corrected Akaike information criterion in jModelTest-2.1.10 [46]. ML reconstruction was performed using the IQ-Tree software (Vienna, Austria) [47]. For branch support, ultrafast bootstrapping [48] was performed with 5000 replicates with 1000 maximum interactions and 0.99 minimum correlation coefficients, and the SH-aLRT branch test was performed with 5000 replicates to validate the ultrafast bootstrapping result. The heuristic search method used was the program’s default, and the algorithm to obtain the final tree was Neighbor Joining. Bayesian inference was performed in the MrBayes program [49,50], using the Bayesian Markov Chain Monte Carlo method to assign trypanosomatids prior to information. Four independent runs were performed for 20 million, with sampling every 2000 generations and 25% burn-in from each run. All the programs were available on the Phylosuite v1.2.2 platform (China) [51] and the reconstruction trees were visualized in FigTree v.1.4.3 software.

A pairwise distance matrix (PDM) was performed for the Eubodonida order to evaluate genetic divergence between different sequences. The Tamura–Nei parameter model, plus gamma distribution among sites (TrN + G), was used. The analysis was performed using MEGA7 software [45].

### 2.6. Statistical Analysis

The performances of the HC and the BC methods were compared by McNemar’s test [52] using the online OMNI Calculator, available in https://www.omnicalculator.com/statistics/mcnemars-test (accessed on 26 October 2022).

The Cohen’s kappa coefficient was calculated to determine the strength of agreement between the HC results and the BC results, which was interpreted as no agreement (0–0.20), minimal agreement (0.21–0.39), weak agreement (0.40–0.59), moderate agreement (0.60–0.79), strong agreement (0.80–0.90), and almost perfect agreement (>0.90) [53]. Two evaluations were performed: (1) the agreement between negative and positive results, independently of the characterized species (*n* = 472); and (2) the agreement between the characterized species of both HC and BC positive results (*n* = 31). Data analyses were performed using the “pacman” and “vcd” packages, implemented in the R platform [54,55]. A *p*-value greater than 5% confidence (*p* < 0.05) was considered significant for all the analyses mentioned above.

## 3. Results

### 3.1. Performance of the First Aliquot and the Sediment Samples

The first aliquot and the sediment samples were demonstrated to be suitable samples for the identification of cultivable and non-cultivable *Trypanosoma* species undetected by the conventional isolation methods in axenic media. We observed two dissimilar profiles resulting from the comparison between the first aliquot and the isolate of 12 bats: the identification of *T. dionisii* and *T. cruzi*, respectively, in the first aliquot and isolate sample. This result revealed mixed infections, and suggested positive selection of *T. cruzi* over *T. dionisii* in the axenic media (Table 3). The use of the sediments allowed for the identification of TcI, *T. dionisii*, *T*. sp. Neobat 1, and *T*. sp. Neobat 4 (Table 4 and Appendix A).

### 3.2. Performance of the HC and BC Techniques

The results of the characterization of kinetoplastids, detected by the HC samples (isolates and sediments) and the BC samples of 472 bats, are exhibited in Table 4 and Appendix A.

The molecular diagnosis of the parasitic DNA extracted from the BC was demonstrated once more to be a valuable strategy for *Trypanosoma* diagnosis. This technique performed better than the HC (χ^2^ = 26.305; *p*-value ˂ 0.0001); it showed 78% of sensitivity (116 positive bats detected through the BC from 149 positive bats detected through the BC and/or the HC) and detected 14 taxa. Meanwhile, the HC showed 41% sensitivity (61/149) and detected 10 taxa.

Nevertheless, the BC method is unable to replace the HC in terms of *Trypanosoma* detection. Instead, the combination of the BC method with the HC considerably enhanced the sensitivity and revealed mixed infections. It was found that 21% of negative bats by the HC were, in fact, positive by the BC. On the other hand, 8% of bats considered negative by the BC were found to be positive by the HC. The Kappa coefficient values for negative and positive results between the characterized trypanosomes were, respectively, 0.22 and 0.26 (*p* values < 0.05), indicating minimal agreement between these methods.

### 3.3. Identification of a New MOTU of the Genus Bodo

We detected a new MOTU of the order Eubodonida in one *Artibeus lituratus* (Stenodermatinae, Phyllostomidae) captured at the Fiocruz Atlantic Forest Biological Station. The sequence obtained from the BC sample of LBT 10097 (GenBank accession no. OP104266) showed 100% coverage and 91.85% identity (E-value = 0.0) with *Bodo saltans*. A phylogenetic tree based on SSU rDNA, inferred through ML and Bayesian analyses, showed LBT 10097 to be basal to one of the clades constituting the order Eubodonida (Figure 3). The genetic distances of the SSU rDNA locus among representatives of the order Eubodonida ranged from 5.3 to 19.5% (Appendix A).

## 4. Discussion

Mixed infections by different species or genotypes of trypanosomes are a common and puzzling phenomenon [16,34,37]. Thus, it is critical to refine the diagnostic techniques and to understand to what extent these methods detect trypanosome infections. Isolation in culture media for the identification and characterization of these parasites is widely performed, and the BC was recently recognized as an important source of *Trypanosoma* DNA [19]. However, the limitations of both techniques are unknown.

Our results disclose four main aspects: (1) a new MOTU of *Bodo* sp. in the bat species *A. lituratus*; (2) the potential of *T. cruzi* selection in the HC; (3) the advantages of employing the first aliquot and sediment samples for the diagnosis of trypanosomatids; and (4) the combination of the HC and the BC techniques, which enhances the detection of single and mixed trypanosomatid infection.

The very low identity (91.85%) of the MOTU LBT 10097 with its closest related species, *B. saltans*, evidences the fact that the diversity of the eubodonids is highly underestimated [3]. Moreover, the large genetic distance of the extant representatives of the order (Appendix A) brings into question whether it is a novel species.

We ruled out external contamination, since bats were submitted to careful asepsis throughout blood collection, and all procedures in the field and laboratory were performed using disposable and autoclaved materials. Although the idea is mind-blowing, we cannot confirm infection by this undescribed organism, since this bat could have ingested contaminated water, and meal-derived DNA fragments can overcome degradation and enter the bloodstream, as observed in humans [56]. Moreover, even if the DNA originated from living flagellates, it is impossible to ascertain whether the infection would be sustainable. Despite this, we cannot rule out infection by *Bodo* in mammals, and this deserves future studies. Within the class Kinetoplastea, the eubodonids descend from orders that partially or fully include parasitic species [3]. Thus, Eubodonida could also present exceptions in their status of mode of life. Further, other free-living protozoans may cause infection in mammals, as observed in opportunistic amoebae [57].

The culturing in axenic media apparently favors *T. cruzi* over *T. dionisii*. Assuming that PCR and Sanger sequencing are biased towards the most abundant genotype [58,59,60], *T. dionisii* was initially the major population. This fact notwithstanding, TcI outgrew *T. dionisii* after the exponential growth phase. Additionally, the ubiquity of *T. cruzi* isolation in the HC from bats concomitantly infected with other cultivable species is intriguing (Table 4). To our knowledge, this is the first observation of *T. cruzi* selection over other trypanosome species by the conventional culturing method in axenic media.

Considering the low sensitivity of the HC and its selective pressure, any association between a particular trypanosome species or *T. cruzi* genotype and a given host species, an epizootiological scenario, or a clinical picture that was based on the encounter of a single species or genotype in the HC becomes questionable. Additionally, as the HC potentially disregards other coexistent parasites, the mixed infections and their outcomes on the course of the parasitosis are overviewed, which is the possible cause of the old-fashioned “one parasite, one disease” approach. With the improvement of the sensitivity and analytical power of diagnosis techniques, these concepts, such as the specificity and the ecology of many trypanosomatids, are being revisited [19,22,60,61,62,63,64].

Herein, we developed an accessible strategy to improve the sensitivity of the hemoculture. Using samples in the initial growth stage of the flagellates in the HC (first aliquot) enables us to overcome the selective pressure of the culture medium. Additionally, the employment of the pellet of non-grown cultures (sediment) allows for the detection of flagellates which are incapable of growing in the axenic medium. Thus, we strongly suggest including these types of samples, which are cheap and easy to obtain, into the laboratory’s routine. Furthermore, we show that even cultivable species may fail to grow, as demonstrated by the presence of *T. dionisii* and *T. cruzi* DTU TcI in the sediments. This might be explained by an initially low flagellate load in the inoculum, or variation of the growth pattern, according to TcI and *T. dionisii* heterogeneity [65,66].

The molecular diagnosis protocol of each sample type was adjusted according to its distinct nature (Table 2). Since parasitic DNA in the blood clot, the first aliquot, and mainly in the sediment was low, we employed extraction methods which provided high yields of good-quality DNA [19,67]; meanwhile, the DNA from the isolate samples could be extracted with a non-toxic commercial kit. The PCR protocol of the sediments differed from the other sample types for the same reason. By modifying the PCR protocol, we doubled our chances of detecting parasitic DNA.

Our results reject our hypothesis regarding the BC method as the gold standard to detect *Trypanosoma* infection. Instead, they show that combining the HC and the BC techniques increases the capacity for diagnosing both single and mixed trypanosomatid infections. In light of the high sensitivity of the BC method to detect trypanosomatid subpatent infections (negative HC) [19], it was surprising that half of the bats with patent infections (positive HC) showed negative results using the BC method. The DNA extraction is performed by taking 50 µL of the blood clot, which is a subsample of a sample of blood. Moreover, the hemoflagellate cells and the DNA molecules are in suspension in the blood and, thus, heterogeneously distributed in the BC. Hence, the 50 µL subsample may not be representative of the diversity present in the individual host.

Nevertheless, the BC technique displays some advantages over the HC. It shows higher sensitivity; it detects higher *Trypanosoma* richness, including non-cultivable species and new MOTUs [17,19]; and it requires a smaller amount of blood. In this sense, we suggest prioritizing the blood samples for BC collection in epizootiological investigations of hemoflagellates in animals with low blood volume.

The overwhelming majority of *T*. sp. Neobat 4 in *Carollia* suggests the occurrence of ecological constraints restricting *T*. sp. Neobat 4 distribution. Thus, investigations evaluating whether *Carollia* spp. share exclusive traits that might expose them to *T*. sp. Neobat 4 infection are needed. Furthermore, the identification of *T*. sp. Neobat 4 in bats belonging to distinct genera (*Anoura caudifer*, *Glossophaga soricina*, and *Platyrrhinus recifinus*) (Appendix A) disproves the previous hypothesis about its specificity to *Carollia* spp. [17].

This work shows that the limitations of the techniques can bias our conclusions about the ecology of the kinetoplastids. The improvement of the analytical power of diagnostic tools will broaden our knowledge of parasite–host interactions and the diversity within this group. Accordingly, we detected a putative new species of *Bodo* sp. in bats. This finding will help to build knowledge on the diversity of eubodonids, and might challenge what is currently known about the mode of life of this group. We also showed positive selection of *T. cruzi* in the HC, which could be overcome using samples in the very initial growth stage of the HC (first aliquot). Together with the sediments, these sample types enhanced the detection of trypanosomatids. Finally, the combined usage of both BC and HC techniques enhanced the sensitivity of the diagnosis of the kinetoplastids, and revealed mixed infections.

## Figures and Tables

**Figure 1 pathogens-12-00071-f001:**
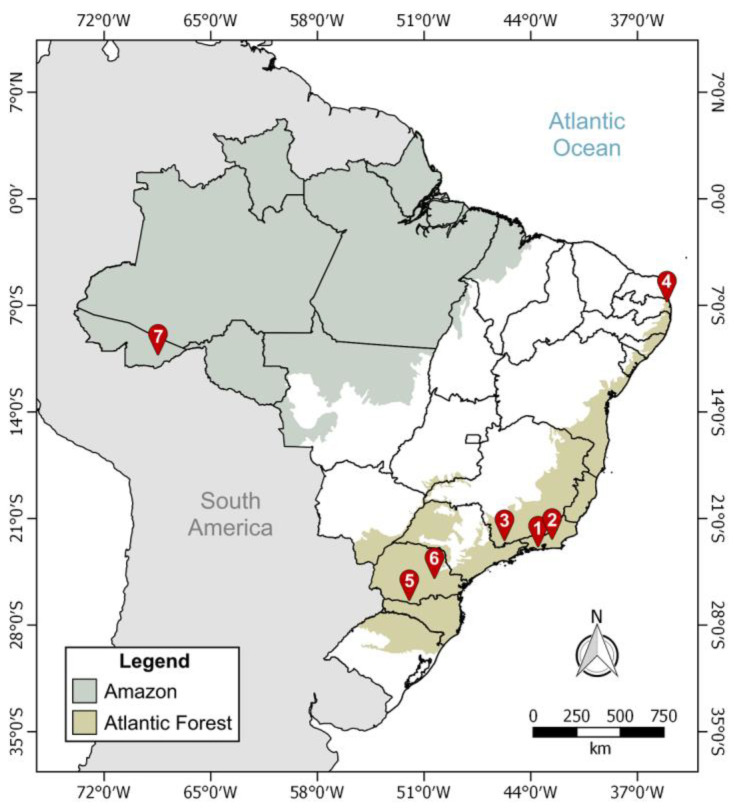
Geographic locations where bats were captured. Bats were collected from 2014 to 2020 at seven Brazilian localities, indicated by the numbered tags: (1) Fiocruz Atlantic Forest Biological Station (Rio de Janeiro municipality, Rio de Janeiro state), (2) Guapiaçu Ecological Reserve (Cachoeiras de Macacu, Rio de Janeiro), (3) Conceição dos Ouros (Conceição dos Ouros, Minas Gerais), (4) Guaribas Biological Reserve (Mamanguape, Paraíba), (5) Palmas Grasslands Wildlife Refuge (Palmas, Paraná), (6) Campos Gerais National Park (Ponta Grossa, Paraná), and (7) Seringal Cachoeira (Xapuri, Acre).

**Figure 2 pathogens-12-00071-f002:**
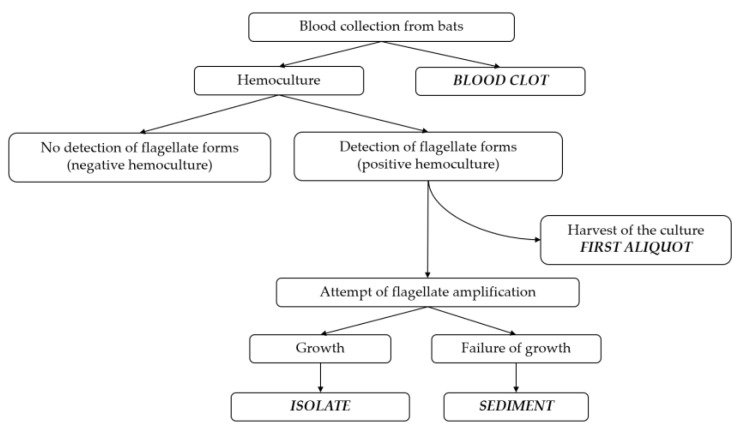
Schematic diagram of the sample collection. The bat blood was distributed for blood clot collection and culturing in biphasic media. There are three sample types from positive hemocultures (i.e., cultures in which flagellate forms have been identified): (1) “first aliquot”: pellet of the culture harvested once the flagellates were detected (i.e., in its very initial growth stage); (2) “isolate”: cryopreserved culture of flagellates that amplified until the stationary phase; (3) “sediment”: pellet of non-grown cultures that showed failure of amplification.

**Figure 3 pathogens-12-00071-f003:**
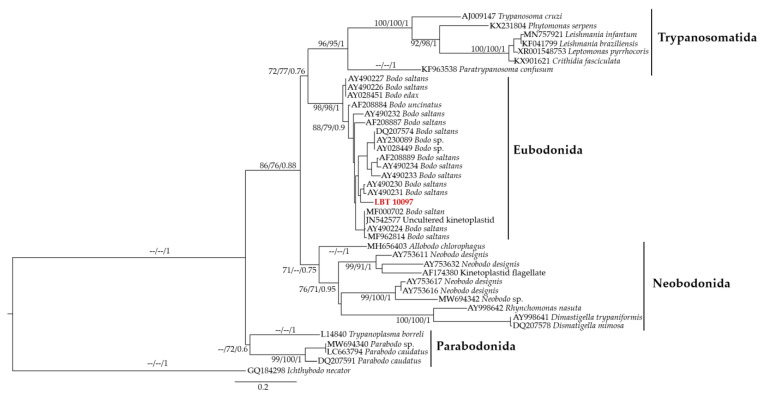
Phylogenetic reconstruction of SSU rDNA sequences from the class Kinetoplastea by maximum likelihood (ML) and Bayesian inference (BI). The analyses were inferred using the TIM3 model with invariant plus gamma distribution among sites (TIM3 + I + G). Maximum likelihood ultrafast, SH-aLRT bootstrap values, and Bayesian posterior probabilities are shown near the nodes, respectively. The scale bar shows the number of nucleotide substitutions per site. The dashes at the nodes represent bootstrap or posterior probability lower than 60/0.6, or branches with congruity in the analysis.

**Table 1 pathogens-12-00071-t001:** Bat samples evaluated in the present study. HC = hemoculture; BC = blood clot.

State	Locality	Positive HC/Total	Excursions (Month/Year)	Source of HC Results	Source of BC Results
Rio de Janeiro	Fiocruz Atlantic Forest Biological Station	9/100	11/2015	[20]	Present data
			03 and 05/2016		
			12/2016	Present data	
			03 and 12/2017		
			09/2018	[21] and present data	
	Guapiaçu Ecological Reserve	12/119	08/2015	Present data	[17]
			10, and 12/2015	[20]	
			02, 04, and 05/2016		
Minas Gerais	Conceição dos Ouros	8/30	07/2019	Present data	Present data
			04/2020		
Paraíba	Guaribas Biological Reserve	3/126	06/2014	[38]	[17]
			03 and 04/2015		
Paraná	Palmas Grasslands Wildlife Refuge	1/30	11/2018	Present data	Present data
			05/2019		
	Campos Gerais National Park	0/18	11/2018	Present data	Present data
Acre	Seringal Cachoeira	28/49	06, 07, and 12/2015	[39]	[17]
Total		61/472			

**Table 2 pathogens-12-00071-t002:** Methodological particularities of the molecular diagnosis for the biological samples.

	First Aliquot	Isolate	Sediment	Blood Clot
DNA extraction method	Phenol-chloroform [40]	DNeasy Blood & Tissue (Qiagen)	Phenol-chloroform [40]	Ammonium acetate precipitation [19]
GoTaq^®^ Green Master Mix (µL)	8.5	8.5	12.5	8.5
PCR primer concentration (µM)	16	16	20	16
PCR product purification	No	Yes	No	No ^1^
Sequencing primer (µM)	16	3.2	20	16

^1^ except for Voucher LBT 7097 (double bands).

**Table 3 pathogens-12-00071-t003:** Comparison of the molecular characterization between the first aliquots and the isolates.

No. of Bats	First Aliquot	Isolate
7	*T. dionisii*	*T. dionisii*
2	TcBat	TcBat
1	*C. mellificae*	*C. mellificae*
1	*T. dionisii*	TcI
1	*T. dionisii*	TcII or TcVI
Total: 12		

**Table 4 pathogens-12-00071-t004:** Results of kinetoplastid detection by the hemoculture (HC) and molecular diagnosis of DNA extracted from the blood clots (BC) of bats. N = number; - = negative result.

HC Result (N)	HC Sample Type (N)	Agreement HC × BC (N)	N	Characterization HC	Characterization BC
Negative (411)	not applicable	yes	326	-	-
		no (85)	41	-	*T*. sp. Neobat 4
			10	-	*T*. sp. Neobat 3
			8	-	*T*. sp. Neobat 1
			5	-	TcI
			5	-	*T. c. marinkellei*
			4	-	*T*. sp. Neobat 2
			2	-	TcIII
			2	-	*T. dionisii*
			2	-	*T. lainsoni*
			2	-	*C. mellificae*
			1	-	TcBat
			1	-	*T. rangeli* A
			1	-	*T*. sp. Neobat 5
			1	-	*Bodo* sp.
Positive (61)	Isolate (34)	yes (12)	9	*T. dionisii*	*T. dionisii*
			2	*T. c. marinkellei*	*T. c. marinkellei*
			1	TcBat	TcBat
		no (22)	5	TcI	*T*. sp. Neobat 4
			5	*T. dionisii*	-
			3	TcI	*T. c. marinkellei*
			3	*T. dionisii*	*T*. sp. Neobat 4
			1	TcI	*T. c. marinkellei* + *T.* sp. Neobat 1 ^1^
			1	TcI	TcBat
			1	TcII or TcVI	*T. dionisii*
			1	TcBat	-
			1	*T. dionisii*	*T*. sp. Neobat 1
			1	*C. mellificae*	-
	Sediment (25)	no	19	TcI	-
			2	*T*. sp. Neobat 1	-
			1	TcIV	*T*. sp. Neobat 4
			1	*T. dionisii*	*T*. sp. Neobat 4
			1	*T. dionisii*	-
			1	*T*. sp. Neobat 4	-
	Isolate and sediment ^2^ (2)	no	2	*T. dionisii* and TcI	*T. dionisii*
				TcIII and *T. dionisii*	*T*. sp. Neobat 3

^1^ Double band sample (Voucher LBT 7097). ^2^ Two bats (Vouchers LBT 6967 and LBT 7148) tested positive in both HC tubes, and flagellate isolation was successful in only one of them. The *Trypanosoma* characterization showed distinct results for each sample type (isolate and sediment).

## Data Availability

The data presented in this study are openly available in the GenBank database (https://www.ncbi.nlm.nih.gov/genbank/) (accessed on 27 October 2022). The SSU rDNA accession numbers are provided in Appendix A.

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
