# Peer review of "Old Methods, New Insights: Reviewing Concepts on the Ecology of Trypanosomatids and Bodo sp. by Improving Conventional Diagnostic Tools"

_pathogens, 2023, doi:10.3390/pathogens12010071_

Round 1

Reviewer 1 Report

Old Methods, New Insights: Reviewing concepts on the Ecology of trypanosomatids and Bodo sp. by Improving Conventional Diagnostic Tools.

Fernanda Moreira Alves et al. present a comparison of haemoculture vs. blood clot to isolate blood kinetoplastid parasites from sylvatic bats by analyzing the amplification of 18S rDNA. They found that a combination of both methods increases the detection of parasitic kinetoplastid fauna in bats. Their work is well written, the experimental design was appropriate for the type of analysis they were meant to do, and the conclusions are derived from the results. In addition, if their approach is applied in other studies, the increased sensitivity in parasite detection will have an impact on the knowledge of parasitic diversity not only in bats but in other sylvatic reservoirs. Therefore their work is suitable for publication, although minor grammar changes that could improve the article.

Line

45. Change “impacts” to “impact”.

63. Change “those” to “these”.

81. Change “commonness” to “high frequency” or “high prevalence” (commonness is more accurate to describe something vulgar rather than something that is frequent).

85. Delete “even less”.

108. Change “assemblage” to “set” or “collection” (assemblage is used more frequently in art terminology).

123. Delete “specimen”.

155. Change “protocol” to “protocols”.

226. Change “in” to “for”.

227. Change “method” to “methods”.

275. Change “infection” to “infections”.

321. Change “kinds” to “kind”.

322 and 351. Change “further” to “furthermore”.

335. Change “potentiates” to “increases”.

341. Change “ul” to “mL”.

345. Change “takes less” to “uses a lower”.

        The methods used to extract DNA are different for each sample type therefore the observed differences might be influenced by this procedure rather than the sample itself (first aliquot and sediment with phenol:chloroform, isolate with a commercial kit, and blood clot with ammonium acetate). The authors should explain why they used different DNA extraction methods for each sample. They should also specify the primers they used for PCR and sequencing reactions, and the cycling protocol for PCR.

Author Response

We would like to express our gratitude for your considerations and your valuable time.

Our detailed point-by-point responses to the comments are given below.

  1. Change “impacts” to “impact”.

The term “impacts” was changed to “impact” (line 46)

  1. Change “those” to “these”.

The term “those” was changed to “these” (line 63)

  1. Change “commonness” to “high frequency” or “high prevalence” (commonness is more accurate to describe something vulgar rather than something that is frequent).

The term “commonness” was changed to “high prevalence” (line 81)

  1. Delete “even less”.

The term “even less” was deleted (line 85)

  1. Change “assemblage” to “set” or “collection” (assemblage is used more frequently in art terminology).

The title of Table 1 was changed to “Bat samples evaluated in the present study” (line 109)

  1. Delete “specimen”.

The term “specimen” was deleted (line 122)

  1. Change “protocol” to “protocols”.

The term “protocol” was deleted due to the inclusion of the primer sequences and cycling protocol for PCR (lines 153-170)

  1. Change “in” to “for”.

The term “in” was changed to “for” (line 232)

  1. Change “method” to “methods”.

The term “method” was changed to “methods” (line 233)

  1. Change “infection” to “infections”.

The term “infection” was changed to “infections” (line 285)

  1. Change “kinds” to “kind”.

We respectfully disagree with this suggestion. We considered the sediment and the first aliquot as distinct kinds of culture samples thus we believe it should be kept in the plural form. Anyhow, to standardize the term we changed “kinds” to “types” (line 331), since the latter is fully applied throughout the text.

322 and 351. Change “further” to “furthermore”.

We agreed and changed to “furthermore” (lines 332 and 361)

  1. Change “potentiates” to “increases”.

We agreed and changed to “increases” (line 345)

  1. Change “ul” to “mL”.

We respectfully disagree with this suggestion. As explained in lines 348 and 349, we take 50 µl of the blood clot for the DNA extraction. In line 351, the subsample refers to the 50 µl blood clot fragment that is taken for the DNA extraction. Anyhow, we corrected “ul” to “µl” (line 351).

  1. Change “takes less” to “uses a lower”.

We changed to “requires a lower” (line 355)

The methods used to extract DNA are different for each sample type therefore the observed differences might be influenced by this procedure rather than the sample itself (first aliquot and sediment with phenol:chloroform, isolate with a commercial kit, and blood clot with ammonium acetate). The authors should explain why they used different DNA extraction methods for each sample. They should also specify the primers they used for PCR and sequencing reactions, and the cycling protocol for PCR.

We appreciate your observation and the explanation why we employed different DNA extraction methods is discussed (lines 336-342). Since parasitic DNA in the blood clot, the first aliquot, and mainly the sediment is low, we employed extraction methods which provide high yields of good quality DNA. For the sediment and the first aliquot, the best option was the phenol-chloroform method (http://dx.doi.org/10.34297/AJBSR.2020.08.001234) and for the blood clot, Rodrigues et al. (https://doi.org/10.1016%2Fj.ijppaw.2019.02.004) found that the DNA extraction based on the ammonium acetate precipitation showed high sensitivity and was suitable for evaluating the diversity of trypanosomes infecting sylvatic mammals, including subpatent and mixed infections. For the extraction of DNA from isolate samples we chose a commercial kit since this method is simple, fast to perform, cost-efficient, produces high quality DNA and has no biohazard issue as the phenol-chloroform method do.

Finally, we specified the primers and cycling protocol in the section “Material and Methods” (lines 153-170).

Reviewer 2 Report

Introduction: Improve the number and the citation of articles.

Material and methods: Improve the table of the dataset. If hard to read especially for the time of the collection.

For line 129 I would suggest making also a digram.

Improve Table 4, because is hard to follow.

Discussion:

Line 290: on what basis do you suggest this hypothesis? 

Author Response

Kind regards,

Fernanda Alves
